# Tany-Seq: Integrated Analysis of the Mouse Tanycyte Transcriptome

**DOI:** 10.3390/cells11091565

**Published:** 2022-05-06

**Authors:** Andrew I. Sullivan, Matthew J. Potthoff, Kyle H. Flippo

**Affiliations:** 1Department of Neuroscience and Pharmacology, College of Medicine, University of Iowa Carver, Iowa City, IA 52242, USA; andrew-sullivan-1@uiowa.edu (A.I.S.); matthew-potthoff@uiowa.edu (M.J.P.); 2Fraternal Order of Eagles Diabetes Research Center, Carver College of Medicine, University of Iowa, Iowa City, IA 52242, USA; 3Iowa Neuroscience Institute, Carver College of Medicine, University of Iowa, Iowa City, IA 52242, USA; 4Department of Veterans Affairs Medical Center, Iowa City, IA 52242, USA

**Keywords:** tanycyte, scRNAseq, neurogenesis, ontology, hypothalamus

## Abstract

The ability to maintain energy homeostasis is necessary for survival. Recently, an emerging role for ependymogial cells, which line the third ventricle in the hypothalamus in the regulation of energy homeostasis, has been appreciated. These cells are called tanycytes and are physically at the interface of brain communication with peripheral organs and have been proposed to mediate the transport of circulating hormones from the third ventricle into the parenchyma of the hypothalamus. Despite the important role tanycytes have been proposed to play in mediating communication from the periphery to the brain, we understand very little about the ontology and function of these cells due to their limited abundance and lack of ability to genetically target this cell population reliably. To overcome these hurdles, we integrated existing hypothalamic single cell RNA sequencing data, focusing on tanycytes, to allow for more in-depth characterization of tanycytic cell types and their putative functions. Overall, we expect this dataset to serve as a resource for the research community.

## 1. Introduction

Whole organism energy homeostasis requires the precise coordination of nutrient intake and energy expenditure. In mammals, energy homeostasis is maintained by the central nervous system through the detection of circulating factors and interoceptive cues that encode relevant biological states and promote a response via endocrine or neuronal efferent signals [1,2,3]. While many areas of the brain constitute the complex network of nuclei that determines energy balance, the hypothalamus has been identified as a hub of regulation, containing numerous regions and cell types integral for homeostatic regulation of food intake, energy expenditure, and blood glucose levels [4,5,6]. Tanycytes are polarized ependymogial cells that line the third ventricle in the hypothalamus. Their cell bodies constitute the wall and floor of the ventricle, while their processes extend deep into the surrounding tissue. The location of tanycytes within the hypothalamus makes them uniquely privileged to sense and respond to energy signals as their apical side is in contact with cerebral spinal fluid and their long processes contact both fenestrated capillaries in the median eminence (ME) and neurons within hypothalamic nuclei that play key roles in the maintenance of energy balance such as the arcuate nucleus (ARH), the ventromedial nucleus (VMH), and the dorsomedial nucleus (DMH) [7,8]. In recent years, tanycytes have drawn considerable attention due to their role in the regulation of food intake, glucose homeostasis, and energy expenditure, all of which contribute to whole-organism energy equilibrium [3]. One tool that has been utilized to investigate the role of tanycytes within energy homeostasis is single cell RNA sequencing (scRNA-seq) [9,10,11,12]. scRNA-seq is a powerful tool capable of revealing meaningful molecular heterogeneity between and within cell types. While many hypothalamic datasets generated from mice containing tanycytes have been produced, this study represents the first effort integrate tanycytic single cell data into a single reference dataset. The integration of single cell transcriptomic data is particularly relevant for tanycytes because tanycytes are generally represented in low abundance in individual data sets. However, by using Seurat integration—a method that allows for an aggregation of transcriptomes from data sets generated across different studies—in this case for four single cell and single nuclei transcriptome hypothalamic datasets--we produced an exclusively tanycyte dataset comprising 14,402 cells. Seurat integration was utilized to control for variability across datasets. Additionally, only samples from mice that were (1) adults at the time of tissue collection, (2) not on a high fat diet, and (3) not engineered or transgenic models were included in this study. Tanycyte isolation and integration workflow can be found in Appendix A. Using this integrated dataset, we were able to tease out subtype specific expression differences of transcripts relevant to tanycyte phenotypes, achieve low *p*-value gene ontology terms for subtypes of tanycytes, compare hypothalamic tanycyte to hindbrain tanycyte-like cells, and investigate the similarities between tanycytes and classical neuronal stem cells.

## 2. Materials and Methods

### 2.1. Hypothalamic Single Cell RNA Sequencing Dataset Transformation, Filtering, and Clustering

Raw hypothalamic single cell RNA sequencing datasets were downloaded from Gene Expression Omnibus (Appendix A). Dr. Guorui Deng was contacted directly to secure the hypothalamic dataset utilized in his 2020 publication [12]. The dataset from Campbell et al. 2017 was downloaded from the website of Dr. Brad Lowell [10]. After loading in the raw data, we used the ‘CreateSeuratObject’ function from the Seurat R package, converting each dataset into a Seurat object. Next, we filtered out cells with a mitochondrial gene percentage greater than 15%, unique features less than 200, and total features below 600. After splitting each dataset based on sample origins, the individual samples of one dataset were integrated together following the Seurat integration pipeline by applying ‘SCTransform’ function to normalize, scale, and find variable features within each sample and used ‘SelectIntegrationFeatures’ for the object list to select 3000 integration features; ‘PrepSCTIntegration’ and ‘FindIntegrationAnchors’ to find features shared between the samples to serve as integration anchors; and, finally, combining the samples back into one dataset again using the ‘IntegrateData’ function.

We next performed principle component analysis on each dataset using the ‘RunPCA’ function and utilized a combination of elbow plots and dimensional heatmaps to determine the appropriate number of principle components for clustering and visualization. We used functions ‘RunUMAP’, ‘FindNeighbors’, and ‘FindClusters’ with resolutions between 0.5 and 0.8 and between 12 and 28 principle component dimensions. The resolution for each dataset was chosen to maximize the visual separation between the clusters on a UMAP.

### 2.2. Cell Population Identification and Integration of Tanycyte Cell Clusters

Using known cell marker genes (Appendix AB,E,H,K), we identified each of the clusters in each dataset, verifying the differential expression with both violin plots and feature plots [9,10,11,12]. After identification, a secondary Seurat object containing only tanycyte clusters from each dataset was created using the ‘Subset’ function. These extracted tanycyte clusters were then merged together using the ‘Merge’ function and stripped of the ScTransform and Integrated assays to allow for reintegration again utilizing the Seurat integration pipeline (see Individual Single cell RNA sequencing dataset transformation, filtering, and clustering). Seventeen PCA dimensions and a resolution of 0.3 was used to cluster the unified tanycytic dataset. Tanycyte clusters were identified using known tanycyte subtype cell markers [9].

### 2.3. Gene Ontology Analysis

G:Profiler (version e104_eg51_p15_3922dba) was used to interrogate GO terms of biological processes enriched in specific cell types using over-representation analysis (ORA). Ensembl mouse genome GRCm39 was used as the reference gene list. Both analyses utilized the ‘FindMarkers’ function, and only returned genes with positive differences between the first identity and second identity to provide us with directional differential expression. Differential expression was determined using a fold change threshold of 0.25 (log-scale) and *p*-value threshold of 1 × 10^−5^.

### 2.4. Hindbrain and Neuronal Progenitor Transformation, Filtering, and Clustering

Raw hindbrain and neuronal progenitor datasets were downloaded from Gene Expression Omnibus (Appendix A). Two hindbrain datasets were used to reach a sufficiently high number of tanycyte-like cells to allow for more meaningful gene ontology analysis. Initially, each dataset was converted into a Seurat object and filtered, removing cells with a mitochondrial gene percentage greater than 15%, less than 200 different mRNA transcripts, and less than 600 total mRNA transcripts. At this point, the two hindbrain datasets were merged together using the “Merge” function. This combined hindbrain dataset was then split by sample origin. The individual samples of these two datasets were integrated together following the Seurat integration pipeline. We utilized the same functions to perform PCA and determine the dimensionality of the hindbrain and dataset as were applied to the hypothalamic dataset. To maximize the visual separation between clusters on the UMAP, a resolution of 0.5 with 35 dimensions was chosen to cluster the combined hindbrain dataset. The cell identity of the resulting clusters was determined using cell markers from their respective publications (Appendix AA,B). The neuronal progenitor dataset was processed through the same sequence of functions and filters as the combined hindbrain dataset. Twelve dimensions and a resolution of 0.5 was used to cluster this dataset after integration (Appendix AC).

### 2.5. Mapping

To map datasets, the Seurat mapping and annotation query datasets vignette was followed. First, to determine cell identities through the recognition of shared features, we applied the ‘FindTransferAnchors’ function after specifying the reference dataset and the query dataset. We then ran ‘TransferData’ to apply the predicted cell identities from the reference dataset to the query dataset and added the identities to the query dataset as metadata using the ‘AddMetaData’ function. Following that, we visually mapped the query dataset onto the reference dataset using the ‘MapQuery’ function. This pipeline was employed to map hypothalamic tanycytes and hindbrain tanycyte-like cells onto the neuronal progenitor dataset as well as the mapping of hypothalamic tanycytes onto the combined hindbrain dataset.

### 2.6. Comparison of Transcript Expression between Tanycyte and Tanycyte-Like-Cells

To compare the relative transcript expression of neuronal stemness markers, neuropeptides, and neuroreceptors between hypothalamic tanycytes and hindbrain tanycyte-like cells, tanycyte-like cells were first extracted from the combined hindbrain dataset using the ‘Subset’ functions. The tanycytes and the tanycyte-like cells were then combined using the ‘Merge’ function and stripped of their non-RNA assays. As these datasets had already been filtered, we next applied the ‘SCTransform’ function to normalize, scale, and find variable features.

### 2.7. Specific Sample Usage and Package Versions

See Appendix A for information on which samples were used from each of the datasets in this study and the version of all packages.

## 3. Results

### 3.1. Integration of Tanycytes and Verification of Identity

By combining four mouse hypothalamic single cell RNA sequencing datasets, this study created a dataset containing 14,402 tanycytes, the largest collection of this cell type to date. Initially, we unbiasedly transformed and clustered each hypothalamic dataset separately using the Seurat Transform pipeline (Appendix A). After identifying all cell clusters using cell specific markers, the tanycyte clusters (*Rax^+^*) from each dataset were extracted. To minimize batch effects and cross-platform variability, we utilized Seurat integration to unify tanycyte populations into a combined dataset (Figure 1A). Next, we clustered the data (Figure 1B) and identified α1, α2, β1, and β2 tanycyte clusters based on previously identified markers for these subtypes (Figure 1C,D) [9]. We also confirmed that each subtype still had high expression for general tanycyte markers *Rax* and *Col23a1* (Figure 1D). To further substantiate molecular classifications assigned to each of the clusters in this dataset, the top ten most differentially expressed genes of each cluster were identified. As shown by the heatmap in Figure 1E, there is substantial heterogeneity of gene expression between these four clusters. Additionally, three of the four subtype markers (*Mafb, Frzb*, and *Scn7a*) are within the top ten most differentially expressed genes for their respective clusters. Lastly, we validated differentially expressed genes between the tanycyte subtypes by cross-referencing the single cell transcript expression pattern against the Allen brain atlas in situ hybridization (ISH) (Appendix A). We investigated both genes highly expressed in individual subtypes and general genes with third ventricle. The gene patterns exhibited a high degree of correlation between single cell transcriptomic and ISH approaches. This further supports the validity of our clustering of tanycytic subtypes within the combined tanycyte dataset.

### 3.2. Transcripts Relevant to Tanycyte Phenotypes Have Subtype Specific Expression

After identifying the specific subsets of tanycytes, we investigated the heterogeneity between the tanycytic subtypes through key gene expression and gene ontology analysis. The purpose of this investigation was two-fold: (1) compare the transcriptional similarities between our combined dataset and previously reported expression; (2) tease out novel markers of functional significance. We started by examining transcripts related to known tanycyte phenotypes such as glucose transport, energy balance, and local regulation of thyroid hormone.

#### 3.2.1. Glucose-Sensing and Energy Balance

Tanycytes are glucose-sensing cells; as such, they express both glucose transporters and glucokinases [13,14]. However, not all glucose transporters are expressed in all subtypes. Previous publications have observed that *Slc2a1* (GLUT1), a major glucose transporter in the mammalian blood–brain barrier, is primarily expressed in β1 tanycytes while β2 tanycytes only express the transporter minimally [15,16]. This is observed in our combined dataset (Figure 2A,B). However, unique to our analysis, there is a similarly low expression of GLUT1 in α1 and α2 subtypes, which suggests that glucose sensing and transport may be mediated by all tanycytes but is possibly primarily controlled by the β1 subtype. Fibroblast growth factor receptors (FGFRs) are another mechanism by which tanycytes respond to energy availability [17]. Work in 2019 by Kaminskas et al. proposes that FGFR1 and FGFR2 present in β tanycytes are the main sites of FGF action on the third ventricle [18]. This distribution of FGFRs is echoed by our dataset’s expression pattern of *Fgfr1 and Fgfr2* transcripts. However, our dataset shows that *Fgfr1*, although expressed more highly in both β tanycyte populations, is primarily enriched in β2 tanycytes (Figure 2A,B). Our dataset also shows heterogeneity in *Fgf10,* a growth factor that marks a population of neurogenic tanycytes that induced an energy balance in regulating neurons [19]. *Fgf10* is expressed in β tanycytes, in agreement with the literature, but within β tanycytes, the β2 subtype has a noticeably higher transcriptional expression of the growth factor (Figure 2A,B). Overall, our integrated analysis provides enhanced insight into the partitioning of glucose sensing and energy balance amongst tanycyte subtypes with β1 tanycytes likely contributing to glucose sensing and transport, whereas β2 tanycytes appear to contribute to energy balance and neurogenesis.

#### 3.2.2. Thyroid Hormone Regulation and Import

Thyroid hormone (T3) is a small tyrosine-based molecule that modulates energy homeostasis by regulating food intake, catabolism in adipose tissue, and neuropeptide Y neuron excitability in the arcuate nucleus [20,21,22]. Local T3 levels in the hypothalamus are regulated by deiodinases (Dio1 and Dio2), which are expressed by tanycytes. Dio2 expression in the third ventricle of the hypothalamus was initially characterized by Tu et al. in 1997 by in situ hybridization [23]. Although Tu acknowledged the idea that subpopulations of tanycytes existed, they were more concerned with rostral to caudal heterogeneity and not the dorsal ventral heterogeneity that defines the subtypes of tanycytes we know today. Thus, while previous publications assert that all tanycytes contain Dio2, our dataset shows that all but α1 tanycytes contain the thyroid transporter (Figure 2A,B). This has important implications as previous studies have used Dio2 as a means to genetically target tanycytes; however, they may have actually accomplished targeting a specific subpopulation of tanycytes [24].

#### 3.2.3. Gene Ontology

In addition to looking at subtype heterogeneity of key transcripts, we employed gene ontology (GO) analysis to uncover population level differences in transcription and function (Figure 2C). The gene transcripts of α1 tanycytes were most highly associated with multiple types of development as well as generation of metabolites and aerobic energy derivation. The α2 subtype transcriptome, on the other hand, was primarily associated with GO terms related to the assembly of cell projections. β1 tanycytes are associated most strongly with the development of GO terms. Finally, the β2 subtype is most associated with gene ontology terms related to anatomical structural development, general development, and responding to organic substances. While each subtype also returned unique GO terms, all subtype transcriptomes were associated with growth: nervous system development, general system development, axoneme or cell projection assembly, or anatomical structure morphogenesis.

### 3.3. Hypothalamic Tanycytes and Purported NTS Tanycyte-Like Cells Lack Similar Transcriptomic Profiles and Are Primed to Fulfil Distinct Biological Functions

Recent work suggests that a tanycyte-like cell exists in the hindbrain, particularly in the nucleus of the solitary tract (NTS) [25]. To determine whether hypothalamic and hindbrain tanycytes exhibit similar transcriptomic profiles and functional markers, we mapped hypothalamic tanycytes onto a combined hindbrain dataset containing a population of tanycytes-like cells (Figure 3A and Appendix AA,B) [25,26]. Most hypothalamic tanycytes localize primarily with astrocytes, endothelial cells, and ependymal cells in the hindbrain (Figure 3B). Considering that tanycyte to astrocyte proliferation has been previously established, endothelial cells share a barrier function with tanycytes and that tanycytes share a similar role as liner of the ventricle; these localizations make biological sense. However, 428 β2 hypothalamic tanycytes mapped onto hindbrain tanycyte-like cells. This tanycytic subtype was also the only type of tanycytes that possessed cells with a similar enough transcriptional profile to map onto tanycyte-like cells. Due to this similarity with tanycytes, albeit only one subtype of tanycyte, we started to investigate other parallels between these two cell types. Hypothalamic tanycytes have been suggested to serve as neuronal progenitors, generating hypothalamic neurons that respond to metabolic cues [19]. To examine whether tanycyte-like cells in the hindbrain are transcriptionally similar to neural progenitors, we mapped both hypothalamic tanycytes and hindbrain tanycyte-like cells onto a neuronal stem cell (NSC) dataset containing cells from the ventricular-subventricular zone (VSVZ) (Figure 3C and Appendix AC) [27]. Approximately 70% of all tanycytes and tanycyte-like cells map onto NSC clusters of the VSVZ dataset (Figure 3D). However, the two cell types primarily map onto different NSC types with hypothalamic tanycytes mapping onto active NSC (aNSC) and tanycyte-like cells from the hindbrain mapping onto quiescent NSC (qNSC). This implicates both hypothalamic tanycytes and hindbrain tanycyte-like cells as neuronal stem cells but with distinct transcriptomes and potentially different propensity for giving rise to newborn neurons. Regardless of NSC clustering differences, this analysis prompted us to investigate the comparative neuronal stemness of hindbrain tanycyte-like cells. Of the seven stemness genes with prominent expression in both hypothalamic tanycytes and hindbrain tanycyte-like cells, the majority had higher expression in tanycytes (Figure 3E). There was, however, one stemness factor transcript, *Erbb4*, that not only had higher expression in hindbrain tanycyte-like cells but was also the gene with the highest overall differential expression between the two cell types. This may contribute to GO terms associated with genes differentially expressed in each of the two cell types, as the top five biological processes upregulated in hindbrain tanycyte-like cells are all related to neuron generation (Figure 3F). This also suggests that the two populations’ stemness are driven by different transcription factors, with hypothalamic tanycytes possibly having a wider range of factors, leading them to neurogenesis, while hindbrain tanycyte-like cell stemness is primarily mediated by *Erbb4*. We next considered the relative expression of neuropeptides and key receptors between hindbrain tanycyte-like cells and hypothalamic tanycytes (Figure 3G). Of the gene transcripts investigated, a substantial number had high differential expressions between the two cell types. This implies that while they occupy a similar space within ventricle structure, the cues that hypothalamic tanycytes and hindbrain tanycyte-like cells respond to and transmit may be quite different.

## 4. Discussion

This study represents an integrated analysis of single cell RNA sequencing data of hypothalamic tanycytes and is the largest collection to date of scRNA-seq data exclusively of this cell type. Few hypothalamic scRNA-seq datasets contain enough tanycytes to tease out subtype heterogeneity. However, an understanding of the subclass intricacies of tanycytes is crucial to our comprehension of their role within the hypothalamus to regulate systemic energy homeostasis. While many tanycytic phenotypes are known, not all tanycytes play equal roles in these pathways, and the hetereogeneity of these cells is only beginning to be understood. The data in this study aids in this elucidation by offering a more refined subtype characterization of transcripts related to glucose sensing, energy balance, and thyroid hormone regulation. Gene ontology analysis further reveals subtype specific differences but also suggests that all tanycytes share a developmental niche as the gene transcripts from all subtypes are associated with GO terms related to development. Furthermore, by interfacing our hypothalamic tanycyte dataset with the tanycyte-like cells of the hindbrain, we were able to detect potential similarities and differences between these cell types. Hypothalamic tanycytes mapped onto a combined hindbrain dataset clustered primarily with astrocytes, endothelial cells, and ependymal cells. However, a fifth of all β2 tanycytes mapped onto hindbrain tanycyte-like cells, suggesting some potential transcription and functional similarities between this subtype and their hind brain counterparts. Both cell types mapped onto neuronal stem cells of the ventricular and subventricular zone, hypothalamic tanycytes to aNSC, and hindbrain tanycyte-like cells to qNSC, pointing toward a shared neuronal stemness ability but potentially different mechanisms. Additionally, tanycytes and tanycyte-like cells express different stemness markers, with tanycytes expressing many at a high level and tanycyte-like cells primarily expressing *Erbb4*. GO analysis revealed that the transcriptome of tanycyte-like cells is more strongly associated with neurogenesis gene terms, further supporting the possibility that this population may serve as a reservoir for neuronal progenitors in the hindbrain. Through the production of the largest collection of tanycytes to date, the true strength of this work is its role as a resource for future investigations into tanycyte subclass phenotypes.

## 5. Limitations

There are several limitations to this study. First, this study is strictly an integrated analysis of transcriptional data. No conclusions about in vivo physiology can be drawn outside of the relative expression of mRNA transcripts. Additionally, while only mice were (1) adults at the time of tissue collection, (2) were not on a high fat diet, and (3) not knockout or overexpression models, there were still age, sex, and background variations between the mice, which could lead to bias within the data. Second, this work generated some compelling hypotheses about intra-tanycytic heterogeneity and similarities and differences between the cell type and tanycyte-like cells. However, while the upmost caution was taken in selecting the specific transformations and analysis tools, multiple unanticipated sources of error could occur from the integration of data collected from different labs using different techniques.

## Figures and Tables

**Figure 1 cells-11-01565-f001:**
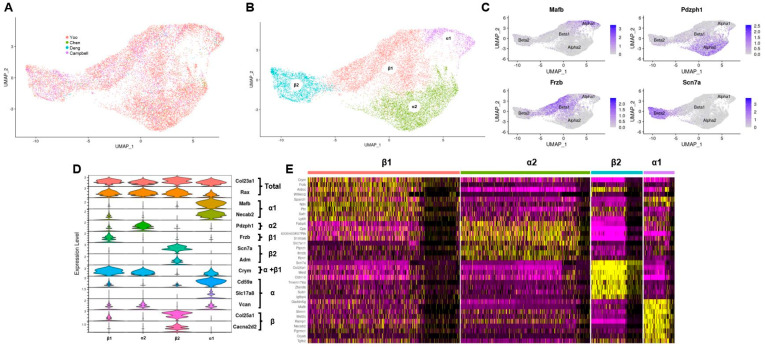
Integration of tanycytes into one dataset. (**A**) UMAP of combined hypothalamic tanycytes split by dataset origin; (**B**) UMAP of combined hypothalamic tanycytes clustered by subtype; (**C**) feature plots of tanycytic subtype specific marker genes identifying the cell clusters of the combined tanycytic dataset; (**D**) violin plots of tanycytic subtype specific marker genes identifying the cell clusters in the combined tanycytic dataset; (**E**) heatmap of the top ten gene transcripts with the highest differential expression in each tanycytic subtype.

**Figure 2 cells-11-01565-f002:**
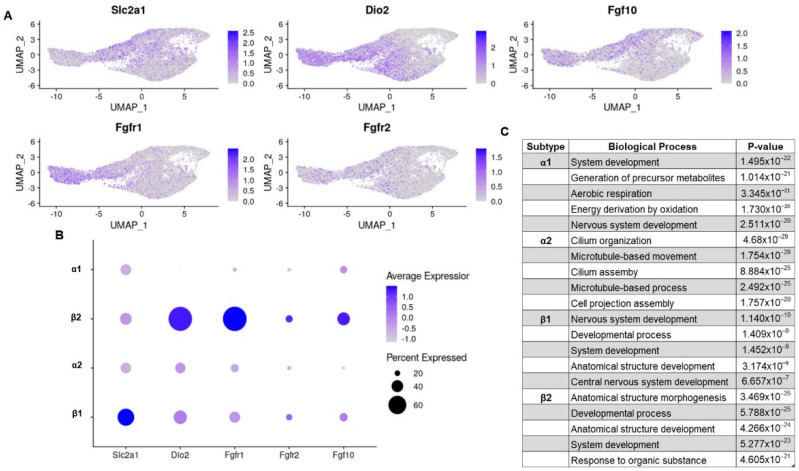
Subtype-specific differential expression of genes related to tanycyte function. (**A**) UMAP feature plots; (**B**) dotplots showing the differential expression of genes related to established tanycytic phenotypes; (**C**) tanycytic subtype specific gene ontology terms of notice.

**Figure 3 cells-11-01565-f003:**
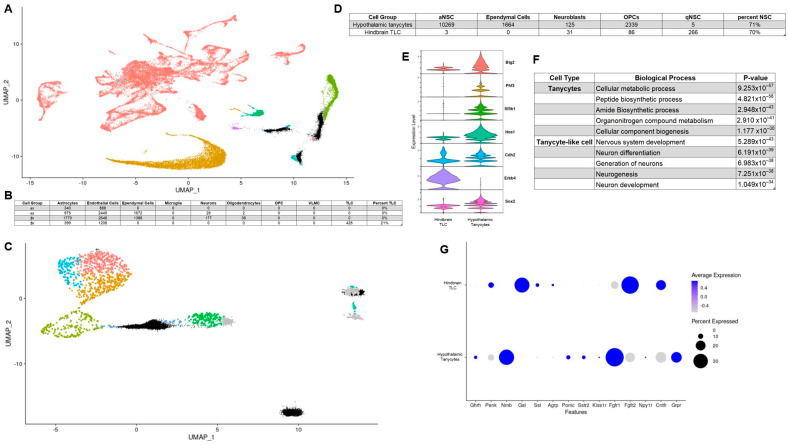
Comparing the expression profile of hypothalamic tanycytes and hindbrain tanycyte-like cells. (**A**) UMAP of hypothalamic tanycytes mapped onto hindbrain dataset (tanycytes in black); (**B**) Number of tanycyte subtypes mapped to each cell type in hindbrain dataset (OPC = Oligodendrocyte progenitor cells; VLMC = vascular and leptomeningeal cells; TLC = Tanycyte-like cells); (**C**) UMAP of hypothalamic tanycytes (black) and hindbrain (grey) tanycyte-like cells mapped onto VSVZ neural stem cells; (**D**) number of tanycytes and tanycyte-like cells mapped to each cell type in NSC dataset. (**E**) violin plots for stemness markers; (**F**) results of gene ontology analysis for terms enriched in tanycytes relative to tanycyte-like cells and vice versa; (**G**) example dot plots of neuropeptide and neuropeptide receptor expression in hypothalamic tanycytes and hindbrain tanycyte-like cells.

## Data Availability

The data presented in this study are openly available in Mendeley Data (reference number pending).

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
