# Peer review of "Tany-Seq: Integrated Analysis of the Mouse Tanycyte Transcriptome"

_cells, 2022, doi:10.3390/cells11091565_

Round 1

Reviewer 1 Report

The authors have taken into consideration the minor comments, and have made an additional analysis in regard to my major comment;  I think you should include a small paragraph which describes your response and refers to Sup Fig 4 data. Also, note that there are no phrases in which you introduce Sup Fig 1 panels (except for panel A), Sup Fig 2, and Sup Fig 3.

Line 143: please change "Supplemental Table 3" for "supplemental Table 2". Lines 333, 334; please introduce a space between words

Reviewer 2 Report

In my opinion, the presented manuscript is interesting and can be a valuable source of new data, but the manuscript requires extensive modification and correction. The authors work only on data received from the database and there is a lack of any other validation such as qPCR. I want to point out that the highest imitation of the research is the fact that the data come from different experiments, probably from animals treated differently, and the RNAs-eq itself was performed on different reagents and on different NGS platforms under different conditions. The bias during the comparison of such data can be significant.

Specific comments:

Tittle - there is no such sequencing method as “Tany-seq”, the method used was RNA-seq or scRNA-seq and please used the appropriate nomenclature

Lines 67-68 – show the number of biological samples used and the numbers of samples in each group

Line 68 – Table S1 – specified the NGS platform and NGS sequencing parameters (especially the cycle numbers)

Line 69 – ‘2020 Hyperthension publication” – what does it mean?

Line 69 – “Campbell et al database” please specify

Line 89 – Which markers? It should be clarified

Lines 100-104 the statistical test used to GO analysis should be presented as well as the reference used

Line 104 – the criteria for classifying genes as differentially expressed should be shown (FC and pvale or FDR)

Line 134 - the criteria for classifying genes as differentially expressed should be shown (FC and p-value or FDR), the statistical approach used

Table S2 – the is a lack of Table S2 footnote

Line 146 - Did you verify the obtained results for gene markers specified for tanycytes using qPCR  (even on different biological material)

Round 2

Reviewer 2 Report

The authors have responded to all of the comments and suggestions and the manuscript has been improved significantly.

I still have doubts about tittle and using the ‘Tany-seq’ naming. The nomenclature RNA-seq; Chip-seq; RIP-seq an so on are the exact NGS approaches used. Creating new names using the names of the cells we use may be confusing the reader. It is not the NGS approach and should be corrected.

I recommend presenting the manuscript for publication after minor modification.

This manuscript is a resubmission of an earlier submission. The following is a list of the peer review reports and author responses from that submission.

Round 1

Reviewer 1 Report

In this paper the authors integrate 4 data bases containing single cell transcriptomic data of C57 mice tanycytes lining the third ventricle ventro lateral walls. Seurat integration pipeline was used. They argue that the integration of multiple data bases will give a more precise description of tanycyte cell type diversity, than data bases of smaller size. They validate their results comparing with the literature and compare the new data base with the transcriptome of hindbrain single cells to test whether it contains tanycyte like cells. This study is relevant since it will give an improved resource for research on tanycytes. However, the validation of the resulting tanycyte data base could be improved.

Major Points:

-The data have been validated by comparing with the literature, but although useful this seems rather limited in scope, since just a few genes were discussed. It should be possible to use the coronal sections at appropriate level from the Allen mouse brain atlas to systematically test the subtype specific predictions.

Minor points:

-It may the help the reader if you explain in the introduction the rationale for using the SEURAT integration.

-Lines 67-69.  Could you please clarify the meaning of “unique and total features”

-The studies from which data were integrated contain multiple types of C57 mice; thus, the authors should comment about the criteria they used to include or exclude animals, and the impact of transgene, sex or developmental stage inclusion on the final result.

Reviewer 2 Report

The manuscript has major problems and the analysis is superficial and not convincing.

For instance it is required to indicate that the presented is the transcriptome of mouse cells.

The following sentence in the introduction, is confusing-

"scRNA-seq datasets contain under 300 tanycytes, making single dataset intra tanycytic heterogeneity difficult to unravel. However, through integration of four single  cell and single nuclei transcriptome hypothalamic datasets, we produced an exclusively tanycyte dataset comprised of 14,402 cells."

How do 4 datasets each with 300 cells give 14,402 cells?

It will be useful to add a flow diagram of the analysis steps done.

The information in Supplemental Figure 1: Identification of cell clusters in hypothalamic reference datasets, does not include the most important information, which are the clusters you subset? Need to add an indication of these clusters high and unique expression of Rax in all of the experiments analyzed and the number of cells identified from each set.

The authors claim identification of the cell types α1, α2, β1, and β2 in the tanycyte clusters based on previously identified markers for these subtypes, they write -“Additionally, three of the four subtype markers (Mafb, Frzb, Scn7a) are within the top ten most differentially expressed genes for their respective clusters.” Moreover, the auther claim that the gene Frzb is β1 specific. Yet, Langlet et al. write “Moreover, α tanycyte markers include Cd59a, Slc17a8, Crym, and Vcan; α2 and β1 tanycyte markers include Frzb and Penk; and β tanycyte markers include Col25a1, Cacna2d2, and Adm gene…”  and Yoo et al (2019) “tanycyte markers such as Rax (α + β), Crym (α + β1), Frzb (α2 + β), and Adm (β2)”

They need to address these contradictions, and show the expression of all known markers of tanycytes cell types in their clusters.